# Moisturizing and Antioxidant Effects of *Artemisia argyi* Essence Liquid in HaCaT Keratinocytes

**DOI:** 10.3390/ijms24076809

**Published:** 2023-04-06

**Authors:** Ziwen Wang, Qiaoli Wang, Wenshen Zhong, Feng Liang, Yuying Guo, Yifei Wang, Zhiping Wang

**Affiliations:** 1Guangdong Provincial Key Laboratory of Advanced Drug Delivery Systems and Guangdong Provincial Engineering Center of Topical Precise Drug Delivery System, School of Pharmacy, Guangdong Pharmaceutical University, Guangzhou 510006, China; 2Department of Cell Biology, College of Life Science and Technology, Jinan University, Guangzhou 510632, China

**Keywords:** *Artemisia argyi*, moisturizing, antioxidant, EGFR, NF-κB, Nrf2/HO-1

## Abstract

*Artemisia argyi* essence liquid (AL) is an aqueous solution extracted from *A. argyi* using CO_2_ supercritical fluid extraction. There have been few investigations on the aqueous solution of *A. argyi* extracted via CO_2_ supercritical fluid extraction. This study aimed to explore the moisturizing and antioxidant effects of AL and to clarify the potential mechanism underlying those effects. Expression levels of skin moisture-related components and the H_2_O_2_-induced oxidative stress responses in human keratinocyte cells were measured via quantitative RT-qPCR, Western blot, and immunofluorescence. Our results showed that AL enhanced the expression of AQP3 and HAS2 by activating the EGFR-mediated STAT3 and MAPK signaling pathways. In addition, AL can play an antioxidant role by inhibiting the NF-κB signaling pathway and activating the Nrf2/HO-1 signaling pathway, consequently increasing the expression of antioxidant enzymes (GPX1, SOD2) and decreasing the production of reactive oxygen species (ROS). This study revealed that AL could be used as a potential moisturizing and antioxidant cosmetic ingredient.

## 1. Introduction

The epidermis, dermis, and subcutaneous layers comprise the human body’s largest organ, the skin [1]. The skin protects the body from environmental damage by covering its exterior surface. Keratinocytes (KCs), which make up the majority of cells in the epidermis, safeguard the skin against damage from the outside world. Water loss from the stratum corneum and keratinocytes’ destruction can lead to skin barrier damage, eventually leading to skin aging [2]. Promoting hyaluronic acid (HA) and aquaporin 3 (AQP3) expression prevents moisture loss from the stratum corneum [3]. There are three primary forms of hyaluronan synthases (HAS1, HAS2, and HAS3), which synthesize hyaluronic acid in the skin [4]. Normally, ROS are essential for the maintenance of a complete cell cycle, which includes cell proliferation, differentiation, migration, and death [5]. However, oxidative stress is caused by a buildup of ROS in epidermal keratinocytes, which will cause dermal–epidermal junction flattening, decreased barrier function, increased transepidermal water loss, and increased pigmentation [6]. Several antioxidant enzymes are produced by cells to regulate ROS production. To maintain a dynamic balance of ROS, the skin primarily relies on the cells to produce more antioxidant enzymes, such as catalase (CAT), glutathione peroxidase (GPx), and superoxide dismutase (SOD) [7,8]. In order to inhibit oxidative stress in KCs, maintaining a balance between ROS generation and antioxidant enzyme activity is critical.

*A. argyi* is a natural herbal medicine with hemostatic and analgesic properties and can treat itchy skin and other diseases [9]. Studies have shown that *A. argyi* contains volatile oil, flavonoids, polysaccharides, phenols, terpenoids, and trace elements [10,11,12]. *A. argyi* essence liquid (AL) is also a kind of *A.argyi* extract, an aqueous solution taken from *A. argyi* leaves via supercritical carbon dioxide extraction, so it is different from the common *A. argyi* extract, and there are few research applications of this material. Compared with water vapor extraction, AL is made under a lower temperature, which can better protect bioactive substances such as flavonoids, polysaccharides, and phenolic components [13]. Our prior research has demonstrated that the primary constituents of AL are terpenoid compounds, with Cineole, L(-)-Camphor, and (-)-Borneol being the primary components [13]. In addition, we found that AL contained a substantial amount of flavonoids, polysaccharides, and polyphenols components (Appendix A).

As water-soluble components of plants, polysaccharides, polyphenols, and glycosides have good water retention properties and antioxidant effects through different pathways [14,15]. Kim et al. found that epigallocatechin Gallate, a phenolic molecule found in green tea, can exert moisturizing and antioxidant activities via stimulating the mitogen-activated protein kinase (MAPK) pathway and increasing the production of HAS2 and other factors [16]. Polyphenols can act on the epidermal growth factor receptor /MAPK (EGFR/MAPK) pathway and janus kinase/signal transducers as well as the activators of transcription (JAK/STAT) signaling pathways [17]. Plant polysaccharides can decrease the content of malonaldehyde and increase the content of glutathione (GSH) via the nuclear factor erythroid-2-related factor 2/heme oxygenase 1 (Nrf2/HO-1) pathway [18].

Combining the material basis of AL with the results of the literature review, we hypothesized that AL has moisturizing and antioxidative effects. Initially, the optimal dose and duration of AL treatment on HaCaT cells (a human keratinocyte cell line) were examined, as was, subsequently, the influence of AL on the expression of moisturizing-related factors (AQP3, HAS2) and their moisturizing mechanism. To evaluate the antioxidant effect of AL, the DPPH free radical-scavenging experiment was used to initially evaluate the antioxidant activity of AL, and the H_2_O_2_-induced HaCaT cell oxidative stress model was applied to detect the influence of AL on the expression of ROS and antioxidant enzymes (GPX1, SOD2). Finally, the antioxidant mechanism of AL was elucidated. We believe that a detailed study of AL will provide theoretical justification for its future use.

## 2. Results

### 2.1. Effects of Different Concentrations and Times of AL in HaCaT Cells

The CCK8 assay assessed the 24 h viability of HaCaT cells treated with different concentrations of AL in order to find the optimal concentrations of AL for HaCaT cells. As shown in Figure 1A, when the concentration of AL was not more than 0.16%, the survival rate of HaCaT was not much reduced. The result showed that AL might inhibit the proliferation of HaCaT at high concentrations (greater than 0.32%). In order to avoid the cytotoxic effects, AL concentrations (0.0025%, 0.005%, 0.01%, and 0.02%) with more than 90% viability were selected for follow-up experiments.

To determine the appropriate incubation time of AL on HaCaT cells, they were incubated with 0.02% AL for 6 h, 12 h, and 24 h, separately. Then, we observed the mRNA expression of AQP3, HAS2, and HAS3 at different times. The results (Figure 1B–D) showed that AL promoted mRNA expression of AQP3, HAS2, and HAS3 in HaCaT cells at 24 h. Therefore, a 24 h period was selected for the follow-up experiments.

### 2.2. Effect of AL on the mRNA Expression of Moisturizing Related Genes in HaCaT Cells

AQP3, HAS2, HAS3, Filaggrin (FLG), Transglutaminase 1 (TGM-1), and tight junction protein 1 (TJP-1) can promote hydration of HaCaT cells. In this study, the effect of AL on the mRNA expression of these targets in HaCaT cells was determined via RT-qPCR. The results (Figure 2) showed that AL could upregulate the mRNA expressions of AQP3, HAS2, HAS3, FLG, TJP-1, and TGM-1 in HaCaT cells compared with the normal group. The results suggested that AL could play a role in moisturizing by activating the mRNA expressions of AQP3, HAS2, HAS3, FLG, TJP-1, and TGM-1.

### 2.3. AL Promoted the Expression of AQP3 and HAS2 in HaCaT Cells

AQP3 and HAS2 proteins are the landmark targets of the moisturizing function. Western blot and immunofluorescence were used to detect the expression of AQP3 and HAS2 in HaCaT cells. Then, quantification analysis of protein bands and fluorescence images was made using ImageJ software. The results (Figure 3A–C) showed that, compared with the normal group, the expression of AQP3 and HAS2 protein bands in the 0.005%, 0.01%, and 0.02% AL groups increased. The fluorescence intensity of AQP3 and HAS2 in the 0.02% AL group was significantly higher than the normal group (Figure 3D–G). The results demonstrated that AL promoted the protein expression of AQP3 and HAS2 in HaCaT cells.

### 2.4. AL Exerted Skin Moisturizing Activities by Upregulating the EGFR Signaling Pathway

As a tyrosine kinase receptor, EGFR can activate a variety of signaling pathways in HaCaT cells, such as STAT3 and MAPK, and is also essential in regulating the expression of HAS2 and AQP3 in epidermal tissue [17]. In this study, we discovered that phosphorylated protein expression of EGFR, STAT3, ERK, and p38 steadily increased with increasing doses of AL (Figure 4A−C). However, JNK phosphorylated protein bands did not change (Figure 4C). Further protein band quantitative analysis showed that compared with the normal group, the phosphorylated protein levels of EGFR, STAT3, ERK, and p38 were significantly upregulated in the 0.01% and 0.02% AL groups (Figure 4D−G). These results suggested that AL could upregulate AQP3 and HAS2 protein expression by promoting the phosphorylation of EGFR, STAT3, ERK, and p38 proteins.

### 2.5. AL Reduced H_2_O_2_-Induced Oxidative Damage in HaCaT Cells

This study was based on an oxidative damage model of HaCaT cells induced by H_2_O_2_ to explore the antioxidant effect of AL. The results showed that AL scavenged DPPH radical dose-dependently, and the 50% effective concentration (EC_50_) value was 7.684% (Figure 5A). The survival of HaCaT cells was reduced after H_2_O_2_ treatment (Figure 5B); however, survival was increased via AL for each treatment within 24 h of treatment (Figure 5E). The ROS production of HaCaT cells reached the maximum when the H_2_O_2_ induction dose was 800 μM (Figure 5C,F). After treatment with AL and vitamin C (Vc) for 24 h, H_2_O_2_-induced ROS production in HaCaT cells was significantly reduced (Figure 5D,G). The results showed that AL could protect HaCaT from oxidative damage caused by H_2_O_2_ and inhibit ROS production.

### 2.6. AL Stimulated the Expression of Antioxidase in HaCaT Cells

It has been reported that SODs, GPxs, and CATs are pivotal in oxidative stress [19]. This study found that H_2_O_2_ significantly downregulated the mRNA expression of CAT, GPX1, and SOD2 in HaCaT cells (Figure 6A–C); however, expression was increased following pre-treatment with AL and Vc. A Western blotting experiment also revealed that H_2_O_2_ could downregulate GPX1 and SOD2 protein expression in HaCaT cells. In contrast, protein expression levels of GPX1 and SOD2 in cells were upregulated or even restored to the normal level after treatment with AL and Vc (Figure 6D–F). The results suggested that AL could play an antioxidant role by stimulating the expression of CAT, GPX1, and SOD2.

### 2.7. AL Reduced H_2_O_2_-Induced Oxidative Stress by Regulating the NF-κB Signaling Pathway

We chose the nuclear factor κB (NF-κB) signaling pathway to examine the specific antioxidant mechanisms of AL. This pathway has been demonstrated to rapidly increase gene expression in inflammatory, immune, and oxidative stress [20]. AL dose-dependently decreased the expression of phosphorylated IKK-β, p65, and AKT in the NF-κB pathway (Figure 7). The results suggested that inhibition of the NF-κB signaling pathway appears to be the mechanism via which AL exerts its antioxidant action.

### 2.8. AL Reduced H_2_O_2_-Induced Oxidative Stress by Regulating the Nrf2/HO-1 Signaling Pathway

Nrf2 is a crucial component of the oxidative stress defense system [20]. HO-1 is an important antioxidant enzyme controlled by Nrf2 and controls the amount of ROS inside cells in response to different stressors. Our study (Figure 8A,B) showed that AL could significantly augment the mRNA expression of Nrf2 and HO-1 in a dose-dependent manner. A Western blot experiment revealed that AL significantly increased the protein expression of Nrf2 and HO-1 (Figure 8C–E). The results suggested that AL attenuated H_2_O_2_-induced oxidative stress by stimulating the Nrf2/HO-1 signaling pathway.

## 3. Discussion

AL has been virtually unexplored as a potential cosmetic ingredient although it has immense potential. Previous studies on *A. argyi* extract have focused on its anti-inflammatory properties. Based on the material basis of AL, this study used the HaCaT cell line to explore its moisturizing and antioxidant effects. HaCaT cells were used since this cell line has similar differentiation properties to keratinocytes. In addition, they have more advantages, as they are easy to culture and are commonly used to evaluate cosmetics [21,22]. This study showed that the administration of AL markedly increased the mRNA and protein expression of AQP3 and HAS2, which are essential for hydration, and that its effects may be achieved by activating EGFR and thus affecting its regulatory pathways. In addition, it was found that the antioxidant activity of AL may result from the regulation of GPX1 and SOD2 via the NF-κB and Nrf2/HO-1 signaling pathways, followed by the suppression of ROS expression.

Maintaining a healthy skin condition cannot be accomplished without sufficient moisture in the skin cells. There are two strategies that prevent stratum corneum water loss: one is to ensure adequate hydration, utilizing natural moisturizing factors such as hyaluronic acid (HA) or delivering sufficient water through AQP3; the other is to form strong crosslinks to play a water-locking role. For example, TJP can form continuous fibers to connect adjacent cells, thereby inhibiting the penetration of macromolecules and slowing down the outflow of water [23,24,25]. HA is a crucial component of human interstitial tissue and has essential physiological functions, such as hydration, extracellular space management, osmotic pressure control, lubrication, and promotion of cell repair [26]. HA is produced by hyaluronan synthases 1, 2, and 3, of which HAS2 is more pivotal in regulating HA [27]. Our study found that AL augments the mRNA expression of HAS2 and HAS3 and dose-dependently promotes HAS2 expression at the protein level. As a member of the water channel protein family, AQP3 has a role in transporting water and glycerol and is expressed in the plasma membrane of normal skin basal epidermal keratin-forming cells, which are closely linked to post-epithelial healing and desiccation syndrome [28,29]. Our results revealed that AL significantly stimulated the synthesis of AQP3 at both the gene and protein levels, which may be associated with the skin hydration effect of AL. Additionally, skin moisture content is regulated by the physical barrier proteins that constitute the skin, such as TGM-1, TJPs, and other moisturizing factors. AL could upregulate the expression of FLG mRNA, TGM-1 mRNA, and TJP-1 mRNA; therefore, we surmised that AL could not only exert a moisturizing effect by promoting hydration, but also further enhance its moisturizing effects by augmenting the skin water-locking function.

The moisturizing effect of AL was investigated based on the expression of AQP3 and HAS2. EGFR is a colossal transmembrane glycoprotein with ligand-inducible tyrosine protein kinase activity. EGFR, primarily expressed in normal skin such as the epidermis, can regulate the differentiation and migration of keratinocytes to the skin surface and plays a crucial role in forming the epidermis [30,31,32]. EGFR can influence the STAT3 signaling pathway [17] and the MAPK signaling pathway [33]. EGFR inhibitor erlotinib was found to reduce AQP3 expression by inhibiting phosphorylation of ERK [34], which leads to deleterious effects such as skin dryness. Expression of HAS2 is also affected by the EGFR signaling pathway [35,36]. Therefore, we explored the connection between AL and the EGFR pathway. Our experiments showed that AL could markedly increase the phosphorylation levels of EGFR, STAT3, ERK, and p38. It was summarized that AL may have a moisturizing effect through promoting AQP3 and HAS2 expression by affecting EGFR and thus regulating the STAT3 and MAPK signaling pathways.

Constant exposure to diverse chemical and physical environmental stimuli causes oxidative stress in the skin [37]. Excessive oxidative stress leads to flattening of the dermal–epidermal junction, reduced barrier function, increased transepidermal water loss, increased pigmentation, and ultimately skin aging [1,6,38]. Therefore, cosmetic ingredients with antioxidant effects are increasingly sought by the market. DPPH is one of the most common methods to determine the free radical-scavenging capacity of substances and has the advantages of stability, rapidity, simplicity, and reliability. Therefore, we first used DPPH for a preliminary exploration of the antioxidant effect of AL, and the results showed that AL has good antioxidant activity. We then tested the antioxidant effect of AL using cells. According to the investigation, oxygen radicals produced by H_2_O_2_ directly promote oxidative stress in human keratin-forming cells, and H_2_O_2_ stimulation of HaCaT cells to induce oxidative stress is one of the most commonly used methods for constructing oxidative stress [30,39]. Based on this, we explored the stimulation concentration of H_2_O_2_ and found that 50% of HaCaT cells survived when the concentration reached 1000 μM. However, AL significantly abated H_2_O_2_-induced cell death. On the other hand, we found that when the H_2_O_2_ concentration was higher, the production of ROS was attenuated due to excessive cell death. Therefore, 800 μM was selected as the subsequent concentration of choice, and the results showed that AL could diminish H_2_O_2_-induced ROS production, thus exerting an antioxidant effect.

Then, we explored the role of AL in antioxidant enzymes. In vivo, the antioxidant enzyme superoxide dismutase catalyzes the degeneration of superoxide radicals to generate H_2_O_2_, which is further eliminated by GPx and CAT [40]. Thus, SOD, GPx, and CAT are the core antioxidant enzymes. SOD is an antioxidant enzyme crucial in removing ROS, particularly superoxide anion radicals, and there are three main isoforms, including SOD2, a mitochondrial matrix protein, a lack of which has been demonstrated to promote skin aging in mice [41]. AL can increase the expression of SOD2, which may be one of the reasons for the antioxidant effect of AL. GPX1 is an antioxidant enzyme in the cytoplasm and mitochondria of mammalian cells, capable of scavenging hydrogen peroxide and soluble lipid hydroperoxides [42]. AL markedly and dose-dependently increased the expression of GPX1. Additionally, AL upregulated the expression of CAT mRNA. In conclusion, AL was capable of reversing the H_2_O_2_-induced damage to antioxidant enzymes SOD2, GPX1, and CAT.

Finally, the antioxidant mechanism of AL was explored. The previous study found that the supercritical extract of *A. argyi* essential oil can act on the NF-κB signaling pathway [43]. NF-κB may affect ROS levels by influencing the expression of antioxidant proteins such as GPX1 and SOD2 [44]. Therefore, whether AL has a regulatory effect on the NF-κB signaling pathway was firstly investigated. The results showed that AL could suppress the phosphorylated expression of IKK, p65, and AKT; thus, the antioxidant effect of AL may be associated with the NF-κB signaling pathway. Nrf2 is a major regulator that regulates redox homeostasis, and HO-1 is a target gene that is pivotal to Nrf2-mediated NF-κB inhibition [45]. Our study showed that AL increased the mRNA and protein levels of Nrf2 and HO-1 expression. Additionally, it has been reported that the NF-κB signaling pathway and Nrf2 signaling pathway have a crosstalk based on molecular crosstalk, and they have antagonistic effects [46]. The present study was also consistent with this finding. In summary, AL may exert antioxidant effects by promoting Nrf2/HO-1 and inhibiting the NF-κB signaling pathway.

## 4. Materials and Methods

### 4.1. Materials

AL was kindly supplied by Guangzhou (Jinan) Biomedical Research and Development Center Co., Ltd. Fetal bovine serum (FBS) and penicillin-streptomycin was obtained from Gibco (New York, NY, USA). Sodium Hyaluronate (NaHA) was purchased from Guangzhou Meiren Technology Co., Ltd. (Guangzhou, China). Hydrogen peroxide (H_2_O_2_) and anhydrous ethanol were supplied by Guangzhou Chemical Reagent Factory (Guangzhou, China). Vitamin C (Vc) was obtained from Guangzhou Sijia Biotechnology Co. Ltd. (Guangzhou, China).

### 4.2. Cell Culture

The American Type Culture Collection (Manassas, VA, USA) was the source for HaCaT cells. HaCaT was cultured in cell culture flasks with 4–5 mL culture medium containing 10% FBS and 1% penicillin-streptomycin. The cell culture flasks were placed in a cell culture incubator containing 5% CO_2_, and the temperature was set at 37 °C.

### 4.3. Quantification of mRNA Levels via Quantitative Real-Time PCR

We used a TRIzol reagent (DP424; Tiangen Biotech Co., Ltd., Beijing, China) to isolate the total RNA in HaCaT cells and a MuLV Reverse Transcriptase Kit (AG11706; ACCURATE Bio, Inc., Changsha, China) to synthesize the complementary DNA (cDNA), following the manufacturer’s instructions. The primers (Appendix A) for quantitative RT-PCR were prepared by Sangon (Shanghai, China). Advanced universal SYBR Green SuperMix (AG11702; ACCURATE Bio, Inc., Changsha, China) and cDNA solution were combined. We used a CFX Connect Real-Time PCR system (Bio-Rad Laboratories, Inc., Hercules, CA, USA) to analyze the mRNA expression level; β-actin acted as the standardizer.

### 4.4. Western Blotting

HaCaT cells were lysed using RIPA Lysis (P0013B, Beyotime, Shanghai, China) containing 1 mM PMSF (T506, Beyotime, Shanghai, China) and 50 μM phosphatase inhibitor cocktail (P1082; Beyotime, Shanghai, China). We completely lysed at a low temperature, then centrifuged the supernatant and collected it. Then, we used the BCA protein assay kit (P0010, Beyotime, Shanghai, China) to measure the protein concentrations. Next, SDS-PAGE was used to separate the protein samples, which were then transferred to a PVDF membrane. Five percent bovine serum albumin was used to block the membrane, followed by overnight incubation at 4 °C with the respective primary antibodies. The primary antibodies used were β-actin (4970s), STAT3 (9139S), p-STAT3 (9131S), ERK (9102S), p-ERK (4695S), P38 (9212S), p-p38 (4511S), JNK (9252S), p-JNK (9255S), IKK-β (3416S), p-IKK-β (2078S), P65 (8242S), p-p65 (3033S), AKT (9272S), p-AKT (13038S), which were purchased from Cell Signaling Technology (Danvers, MA, USA); AQP3 (A2838) from ABclonal (Boston, MA, USA); HAS2 (sc-514737), EGFR (sc-373746), and p-EGFR (sc-377547) from Santa Cruz Biotechnology (Dallas, TX, USA); SOD2 (WL02526), GPX1 (WL02497a), Nrf2 (WL02135), and HO-1 (WL02400) from Wanleibio (Shenyang, China). The next day, after washing, the membrane was treated with a secondary antibody (ZEN BIO, Chengdu, China). coupled with horseradish peroxidase for 1 h at room temperature for luminescent imaging (Tannon, Shanghai, China). Protein expression was assessed using ImageJ V1.8.0. software (Bio-Rad, Hercules, CA, USA). β-actin was used to normalize protein expression levels.

### 4.5. Immunofluorescence

We used 4% paraformaldehyde to fix the HaCaT cells for 15 min. They were washed and treated with 0.1% Triton X-100 for 2–20 min, washed again, and then 5% bovine serum albumin was used for blocking for 1 h at room temperature, followed by incubation overnight at 4 °C. We added FITC-conjugated fluorescent secondary antibody (A11005/A21206, Thermo Fisher Scientific, Waltham, MA, USA) the next day and incubated for 1 h. When the primary antibody is HAS2, phalloidin staining is required for 2 h. Finally, we added 0.5 g/mL of DAPI staining solution and kept the cells in the dark for 10 min. After aspirating the dye, we added 1mL of PBS buffer and observed the image with a fluorescent microscope (LSM900, Zeiss, Germany). Using ImageJ software, we analyzed protein expression levels.

### 4.6. DPPH Decolorimetric Assay

The AL was diluted with anhydrous ethanol to a concentration between 0.7% to 50%. Following this, we added it to a 96-well plate (Sample group: 100 μL AL sample + 100 μL DPPH (Yuanye Biotechnology Co., Ltd., Shanghai, China); Blank group: 100 μL anhydrous ethanol + 100 μL DPPH), then mixed thoroughly, kept the mixture in the dark for 30 min, and measured the absorbance at 517 nm using an enzyme marker. Vitamin C (Vc) was employed as the positive control, while the blank group consisted of anhydrous ethanol in place of the AL sample. We then calculated SA (%) according to the following expression and subsequently calculated the EC_50_ value of AL.
%SA = [1 − (A_blank_ − A_sample_)/A_blank_] × 100

A_Samples_ (or A_blanks_) are DPPH treated with AL (or ethanol absolute) at 517 nm.

### 4.7. CCK8 Assay

A DMEM medium was used to dilute the AL to give concentrations from 0.00125% to 0.64%. After 24 h in an incubator containing 5% CO_2_ at 37 °C, we evaluated cell viability using the CCK8 (CK04; Dojindo, Kumamoto, Japan) technique. In 96-well plates, 1.5 × 10^4^ cells/well of HaCaT cells were seeded. After 24 h of treatment with media containing 0.005%, 0.01%, and 0.02% AL, cells were washed with PBS. DMEM media with 1000 µM H_2_O_2_ were added to each well, and the CCK8 test was used to evaluate cell viability at 450 nm after 4 h, with an enzyme immunoassayreader (Bio-Rad Laboratories, Inc., Hercules, CA, USA).

### 4.8. Intracellular Reactive Oxygen Species (ROS) Assay

In a 6-well plate, 4 × 10^5^ cells/well of HaCaT cells were seeded. Additionally, they were treated with AL (0.005%, 0.01%, and 0.02%) or Vc (400 μM). After 24 h, we washed with PBS and co-incubated with a DMEM medium containing 800 μM H_2_O_2_ for 4 h. After washing 3 times with a PBS buffer, 10 μM DCFH-DA(S0033M, Beyotime, Shanghai, China) prepared in a serum-free DMEM medium was added, followed by incubation at 37 °C for 20 min in the dark. The excitation wavelength was 488 nm. We used a fluorescence microscope (Nikon Corporation, Tokyo, Japan) to examine and photograph the samples, and the amount of reactive oxygen species were measured with ImageJ V1.8.0 software.

### 4.9. Statistical Analysis

The means and standard deviations (SDs) of at least three separate experiments were used to express experimental results. For statistical comparisons, the one-way analysis of variance (ANOVA) was employed. Statistical significance was set at *p*-values < 0.05.

## 5. Conclusions

The current study found that AL treatment elevated the expression of moisturizing-related factors such as AQP3 and HAS2 in HaCaT cells by activating the EGFR/STAT3 and EGFR/MAPK signaling pathways. The findings of the DPPH experiment revealed that AL has free radical-scavenging activity. AL pretreatment can inhibit the NF-κB signaling pathway and activate the Nrf2/HO-1 signaling pathway, resulting in the activation of antioxidant enzymes, a reduction in ROS generation, and a weakening of the oxidative stress response of HaCaT cells caused by H_2_O_2_. In conclusion, our research has demonstrated that AL has moisturizing and antioxidant properties, and could be used as a potential cosmetic ingredient.

## Figures and Tables

**Figure 1 ijms-24-06809-f001:**
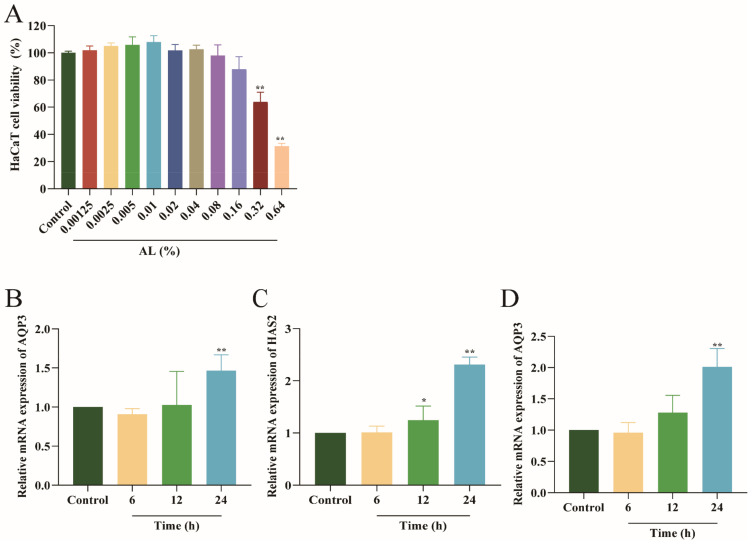
Effects of different AL treatment concentrations and times on HaCaT cells. (**A**) HaCaT cells treated with AL (0.00125~0.64%) for 24 h. The mRNA expression of AQP3 (**B**), HAS2 (**C**), and HAS3 (**D**) in HaCaT cells treated with 0.02% AL for different times. Results are expressed as means ± SDs of three independent experiments. ^**^
*p* < 0.01, ^*^
*p* < 0.05 compared to the normal group (no treatment). One-way ANOVA was used to evaluate significance.

**Figure 2 ijms-24-06809-f002:**
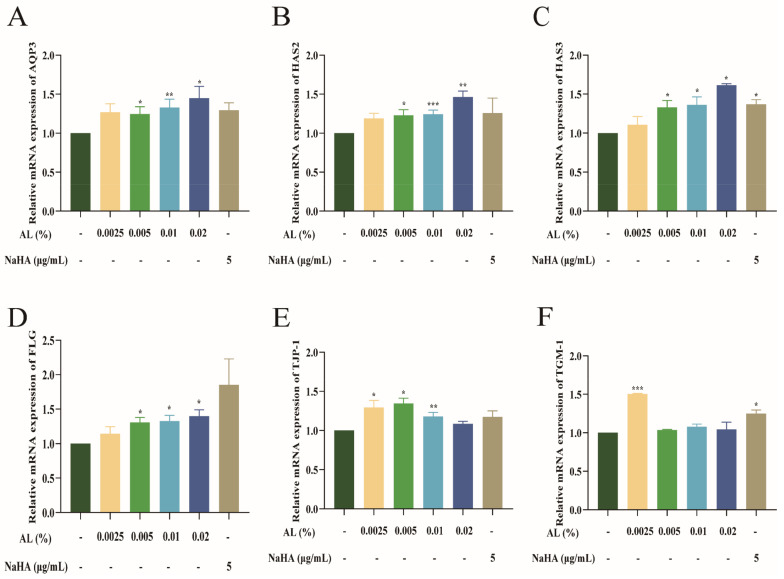
Effect of AL on the mRNA expression of moisturizing-related genes in HaCaT cells. Relative expressions of AQP3 **(A**), HAS2 (**B**), HAS3 (**C**), FLG (**D**), TJP−1 (**E**), and TGM−1 (**F**) were detected in HaCaT cells via RT-qPCR. Results are expressed as means ± SDs of three independent experiments. Relative expression quantity values were calculated relative to the β-actin gene. ^***^
*p* < 0.001, ^**^
*p* < 0.01, ^*^
*p* < 0.05 compared to the normal group (no treatment). One-way ANOVA was used to evaluate significance.

**Figure 3 ijms-24-06809-f003:**
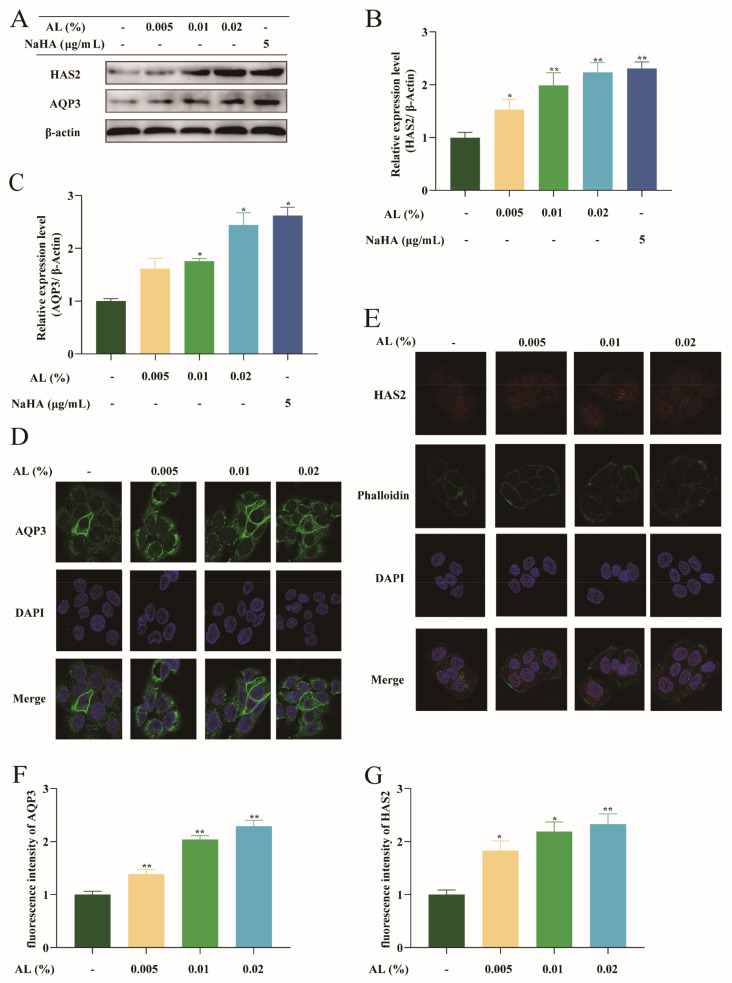
AL activated AQP3 and HAS2 in HaCaT cells. (**A**) Protein bands of AQP3 and HAS2 via Western blotting; ImageJ software protein quantification assay of HAS2 (**B**); AQP3 (**C**) protein expression levels of HAS2 using ImageJ software; (**D**) IF staining of AQP3 in HaCaT cells (63×) and (**F**) its protein quantification analysis; (**E**) IF staining of HAS2 in HaCaT cells (63×) and its protein quantification analysis (**G**). The target protein’s quantitative result is the gray value of the β-actin. Results are expressed as means ± SDs of three independent experiments. ^**^
*p* < 0.01, ^*^
*p* < 0.05 compared to the normal group (no treatment). Significance was measured using a one-way ANOVA.

**Figure 4 ijms-24-06809-f004:**
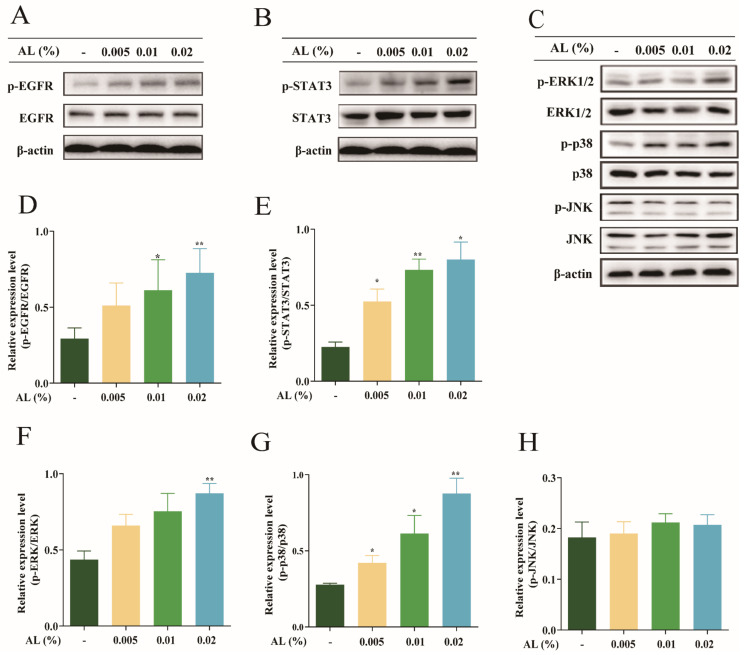
AL exerted skin moisturizing effects by upregulating EGFR signal pathways. Protein bands of EGFR (**A**), STAT3 (**B**), ERK, p38, JNK (**C**) in HaCaT cells; protein quantification analysis of EGFR (**D**), STAT3 (**E**), ERK (**F**), P38 (**G**), JNK (**H**). Results are expressed as means ± SDs of three independent experiments. ^**^
*p* < 0.01, ^*^
*p* < 0.05 compared to the normal group (no treatment). One-way ANOVA was used to evaluate significance.

**Figure 5 ijms-24-06809-f005:**
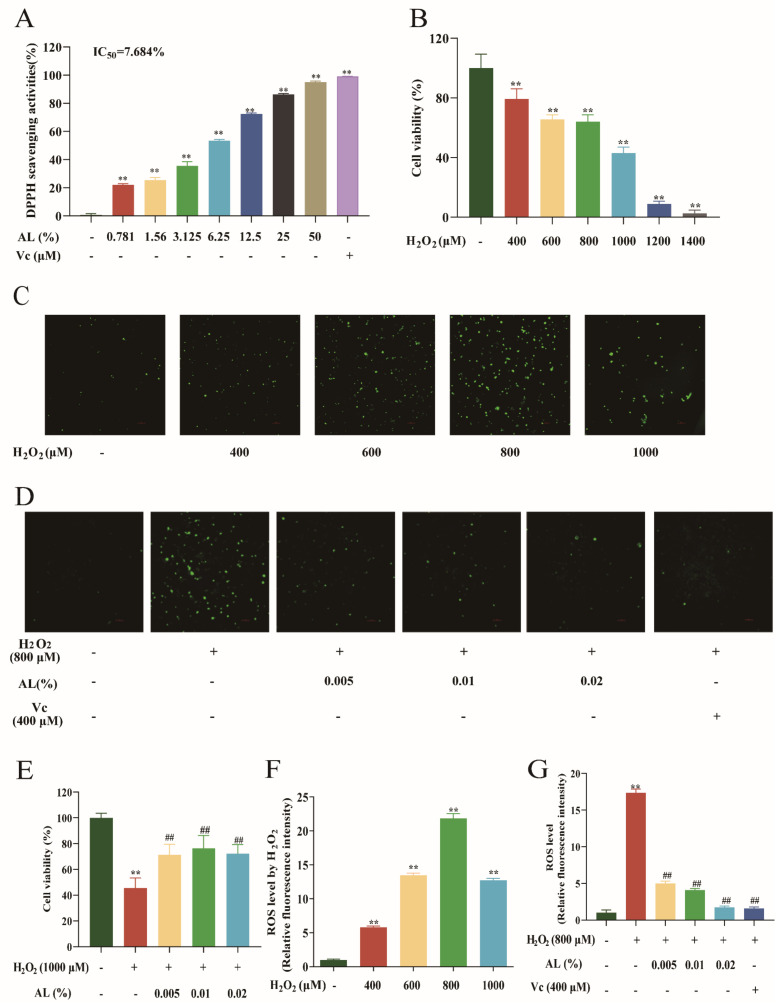
AL reduced H_2_O_2_-induced oxidative damage and ROS in HaCaT cells. (**A**) DPPH Scavenging activities of AL in vitro. (**B**) Cell viability for 24 h after treatment with H_2_O_2_ for 4 h in HaCaT cells. After 4 h treatment with different concentrations of H_2_O_2_ (**C**), or after 24 h treatment with 0.005%, 0.01%, and 0.02% AL or 400 μM Vc (**D**), the HaCaT cells were fluorescently labeled with a DCFH-DA probe, and a representative image of ROS was selected (10×). (**E**) Cell viability for 24 h after treatment with 0.005%, 0.01%, and 0.02%AL or 400 μM Vc using a CCK8 assay. (**F**) Quantitative analysis of ROS images induced by H_2_O_2_ for 4 h. (**G**) Quantitative analysis of ROS images induced by H_2_O_2_ after treatment with 0.005%, 0.01% and 0.02% AL or 400 μM Vc for 24 h. Results are expressed as means ± SDs of three independent experiments. ^**^
*p* < 0.01compared to the normal group (no treatment); ^##^
*p* < 0.01compared with the H_2_O_2_ group. One−way ANOVA was used to evaluate significance.

**Figure 6 ijms-24-06809-f006:**
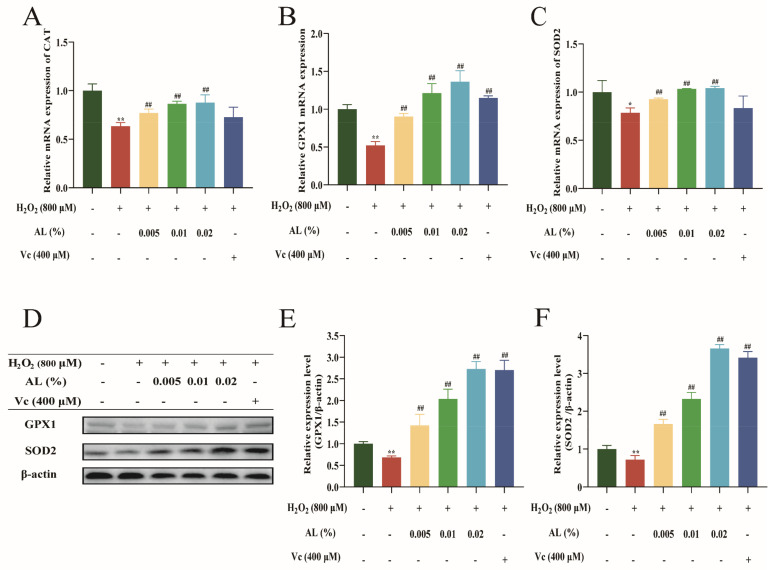
AL stimulated the expression of antioxidases in HaCaT cells. Relative mRNA expressions of CAT (**A**), GPX1 (**B**), and SOD2 (**C**) were detected in HaCaT cells via RT−qPCR. (**D**) Protein bands of GPX1 and SOD2 using Western blotting; protein quantification assay of GPX1 (**E**) and SOD2 (**F**) using ImageJ software. Results are expressed as means ± SDs of three independent experiments. ^**^
*p* < 0.01 compared to the normal group (no treatment); ^##^
*p* < 0.01 compared with the H_2_O_2_ group. One-way ANOVA was used to evaluate significance.

**Figure 7 ijms-24-06809-f007:**
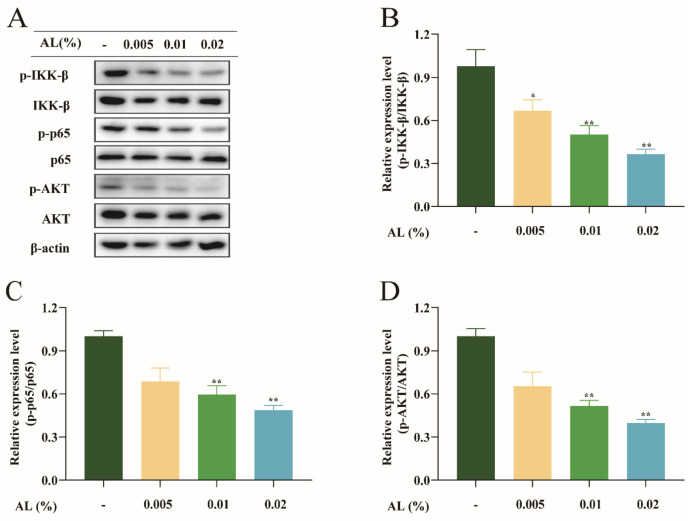
AL reduced H_2_O_2_-induced oxidative stress by inhibiting the NF-κB signaling pathway in HaCaT cells. (**A**) Protein bands of IKK-β, p65, and AKT using Western blotting; ImageJ software protein quantification assay of IKK-β (**B**), p65 (**C**), and AKT (**D**). Results are expressed as means ± SDs of three independent experiments. ^**^
*p* < 0.01, ^*^
*p* < 0.05 compared to the normal group (no treatment). Significance was measured using a one-way ANOVA.

**Figure 8 ijms-24-06809-f008:**
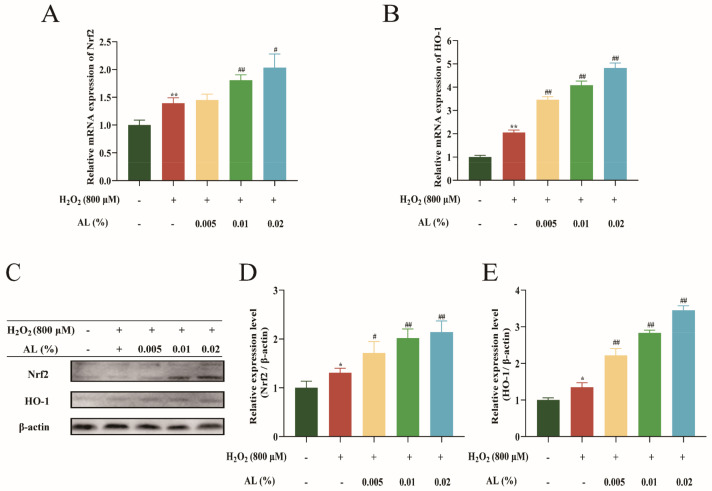
AL reduced H_2_O_2_-induced oxidative stress by promoting Nrf2/HO-1 in HaCaT cells. Relative mRNA expression of Nrf2 (**A**) and HO-1 (**B**) were detected in HaCaT cells via RT-qPCR. (**C**) Protein bands of Nrf2 and HO−1 using Western blotting; protein quantification assay of Nrf2 (**D**), HO-1 (**E**) using ImageJ software. Results are expressed as means ± SDs of three independent experiments. ^**^
*p* < 0.01, ^*^
*p* < 0.05 compared to the normal group (no treatment); ^##^
*p* < 0.01, ^#^
*p* < 0.05 compared with the H_2_O_2_ group. One-way ANOVA was used to evaluate significance.

## Data Availability

The original contributions given in the study are included in the article/Appendix A; the relevant authors can be contacted for further information.

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
