# Peer review of "Moisturizing and Antioxidant Effects of Artemisia argyi Essence Liquid in HaCaT Keratinocytes"

_ijms, 2023, doi:10.3390/ijms24076809_

Round 1

Reviewer 1 Report

This is an original article about the antioxidant effects of Artemisia argyi essence in HaCat keratinocytes. They showed a very clear effect of Artemisia angyi in vitro experiments.  

Artemisia angyi has been used for folk remedy almost all over the world and is considered one of the panaceas in Asian countries since the famous Chinese medical doctor in the late Han dynasty era, Huà Tuó, used Artemisia argyi very often and wrote in his text. Artemisia agyi has been focused on because it is expected to have promising effects on many kinds of diseases same as other herbal medicine. However, some doctors consider that these herbal medicines are probably very fishy for practical indication; therefore, more evidence was required for future production. This manuscript can act as one of the crucial parts to fill the gap of evidence and effects.  

The original aspects of this manuscript in experimental design were 1) they compared the effects of Artemisia argui and Vitamine C, 2) they also considered the factor of moisture in vitro assay. The experimental methods performed in this manuscript are prevalent, but it doesn’t matter.  

This manuscript adds more scientific evidence of Artemisia argyi for wound repair and skin care. Moreover, it can be suggested that Artemisia angyi would be possible future products for skin wound healing and skin care, which probably expand expectations especially in the cosmetic industries.  

Moreover, they clarified the mechanism of Artemisia argyi effects on keratocyte, showed radical scavenging effects and inhibition of H2O2-induced cell death by reducing ROS production, and its clear pathway was promotion of SOD2 and GPX1 expression by Nrf2/HO-1 expression and NF-kappa B signalling pathway inhibition. Artemisia argui has been reported to negatively affect NF-kappa B pathway using cancer cells, however, normal cells such as keratinocytes or fibroblasts were not experienced before. Therefore, as far as I know, this manuscript showed the detail mechanism of Artemisia argyi in normal cells.  

These are in vitro experiments; therefore, further investigation in vivo is definitely required, and they are planning it as the next step.  

Regarding manuscript structure, they are well considered the experimental design, adequately described the results and methods, and the discussion part is consistent with the results. All figures are well-described and easy to understand. I want to say I prefer colourful figures using bright red, yellow, blue and orange, which is just my personal preference. If other reviewers consider the figures should be more colorful, I strongly recommend they switch to colourful and bright figures.
In conclusion, I think this manuscript is suitable for publication from IJMS with this version

Author Response

Dear reviewer,

Thank you for reviewing our manuscript and for the constructive comments, which greatly helped us to improve the manuscript. We have carefully considered your suggestion and made some changes. So we have revised the figures and they are colorful and bright now. All changes made are marked with red color and underlined. We hope that your comments have been addressed accurately. 

Best Regards.

Reviewer 2 Report

The basic researches help us to understand the effects seen in patients.

The cosmetic industry always seeks for new plant ingidients.

The article is well written.

Author Response

Dear Reviewer,

Thank you so much for handling the review of our manuscript. I really appreciate all your comments and suggestion! Thanks again!

Best regards

Reviewer 3 Report

Dear Editor and authors

I write related to "Moisturizing and antioxidant effects of Artemisia argyi Essence Liquid in HaCaT Keratinocytes"

The similarity rate for this article except references was 34%. This rate is very high and not acceptable. Therefore, I have negative doubts about the study. All paragraphs under headings 4.6, 4.7 and 4.8 in this article were copied and pasted directly from an article which is “Antioxidant Activities and Protective Effects of Dendropachol, a New Bisbibenzyl Compound from Dendrobium pachyglossum, on Hydrogen Peroxide-Induced Oxidative Stress in HaCaT Keratinocytes”

The species name should be written italic in the title.

Abstract flow is not good. “We measured the……………… Furthermore we investigate……………..We also evaluate……….In addition we explored…………..Finally we also investigate……………Our result showed………..”

keywords usually consist of one word. I think a four-word keyword is not appropriate.

Kim et al. should be written instead of Eunji Kim.

Survival rates were 125%, 126%, 128%???? a greater than one hundred percent rate.

Legend of axis y in figure 1A contrl???

HaCaT cells treated with hyaluronic acid 5µg/mL. But in figure 3A what is NaHa abbreviation?

I could not see figure legend 5. Figure 4 written twice. It should be corrected.

Legend of axis y in figure 5B vialibility control????

After the species name is written once in the text, the species name can be written short. For example, A. argyi should be written in discussion second sentence.

Corrections:

at different times

Different treatment concentrations

The target protein’s quantitative result is the gray

With the H2O2 group

regulating the NF-κB

to the normal group

an a target

our current research suggests

Author Response

Dear Reviewer,

Thank your careful work regarding our manuscript and for the constructive comments, which greatly helped us to improve the manuscript. We have revised the manuscript in accordance with your comments. All changes made are marked with red color and underlined.  

Point-by-point responses to the comments were as follows:

Question 1: “The similarity rate for this article except references was 34%.”

Response: we have revised the our manuscript. The similarity rate for this article except references was less than 15% now.

Question 2: “survival rates were 125%, 126%, 128%???? a greater than one hundred percent rate.”

Response:compared to the control group, the number of cells increased in the low concentration group. So survival rates of the low concentration group are greater than one hundred percent rate.

We hope that your comments have been addressed accurately. 

Best regards.

Round 2

Reviewer 3 Report

Dear Editor and Authors,

The authors have almost made the requested changes. The similarity rate of the article has also been reduced. However, i suggest some correction listed below.

“ Compare with water vapor extraction, AL was made under has a lower temperature,..” should be rearranged

“…antioxidant effect through different pathways” effects should be written

“the 50% cytotoxic concentrations…” concentration should be written

“..compared with normal group..” the should be added before normal

“…..however them were increased following…” they should be written instead of them

“…by regulating NF-κB signaling pathway.” the should be added before NF-κB

“The results suggests that inhibition” suggest should be written

A. argyi essential oil, can act on..” no need to comma between oil and can

“signaling pathway was firstly investigared”  investigared is miswritten word

“Fetal bovine serum (FBS) and penicillinstreptomycin was obtained” should be written as “Fetal bovine serum (FBS) and penicillin-streptomycin were obtained”

“in a cell culture flasks” should be written a “in cell culture flasks”

“10% FBS and 1% penicillin-streptomycin”

“The cell culture flasks was placed in a cell culture” were should be written instead of was

“….analyze the mRNA expression level and β-actin act as the standardize.” should be rearranged.  Standardize?????

“…were purched from Cell Signaling….” purchased should be written.

“… let stand in the.”””””????

The Supplementary Material for this article can be found online at.” at????

Author Response

Dear reviewer,

Thank your careful work regarding our manuscript and for the constructive comments, which greatly helped us to improve the manuscript. We have revised the manuscript in accordance with your comments. We have polished our manuscript by MDPI’s editing services. All changes made are underlined and marked with blue, red, and purple colors, since two MDPI’s editors and we polished and revised the article. The changes were presented in revision pattern.

Question 1: “ ‘The Supplementary Material for this article can be found online at.’ ”.

Response: specific website address will be added after the manuscript is accepted. 

Best regards.